

# Strategies for Incorporating Static Features into Global Deep Learning Models

Tanja Liesch[1] and Marc Ohmer[1]

[1]Institute for Applied Geosciences (AGW), Karlsruhe Institute of Technology (KIT), Karlsruhe, Germany

**Correspondence:** Tanja Liesch (tanja.liesch@kit.edu)

**Abstract.** Global deep learning (DL) models are increasingly used in hydrology and hydrogeology to model time series data across multiple sites simultaneously. To account for site-specific behavior, static input features are commonly included in these models. Although the method of integration of static features into model architectures can influence performance, this aspect is seldom systematically evaluated. In this study, we systematically compare four strategies for incorporating static features into a global DL model for groundwater level prediction, including approaches commonly used in water science (repetition, concatenation) and two adopted from related disciplines (attention, conditional initialization). The models are evaluated using a large-scale groundwater dataset from Germany, tested under both in-sample (temporal generalization) and out-of-sample (spatiotemporal generalization) settings, and with both environmental and time-series-derived static features.

Our results show that all integration methods perform rather similar in terms of average metrics, though their performance varies across wells and settings. The repetition approach achieves slightly better overall performance but is computationally inefficient due to the redundant replication of static features. Therefore, it may be worthwhile to explore alternative integration strategies that can offer comparable results with lower computational cost. Importantly, the choice of integration method becomes less critical than the quality of the static features themselves. These findings underscore the importance of careful feature selection and provide practical guidance for the design of global deep learning models in hydrologic applications.

## 1 Introduction

In recent years, so-called global or regional deep learning models have gained increasing popularity in hydrology (Kratzert et al., 2018, 2019) and related fields such as hydrogeology (Clark et al., 2022; Heudorfer et al., 2023). In contrast to traditional local models, i.e., models trained on individual basins or wells, these approaches leverage multiple time series simultaneously within a single model. This has two key advantages: First, global models are, at least in theory, capable of generalizing to ungauged basins or sites (e.g., generating spatially continuous groundwater level surfaces). Second, it has been shown that they can achieve superior performance not only in terms of average metrics (Kratzert et al., 2024), but also in predicting extreme events (Frame et al., 2022; Kratzert et al., 2024).

To account for the fact that each unit (e.g., a basin or groundwater observation well) may respond differently to the same dynamic inputs (such as meteorological drivers), global models typically incorporate a set of static input features that describe the unique properties of each unit. Without such static features, a global model can only learn an average response to the





dynamic inputs, which often results in reduced predictive performance (Kratzert et al., 2019; Heudorfer et al., 2023). These static features commonly include characteristics that influence how the output variable (e.g., surface runoff or groundwater level) reacts to meteorological forcing, such as land use, topography, soil type, or geological and aquifer properties.

From a technical perspective, it is not trivial how dynamic and static inputs should be jointly processed in a global deep learning model. While several options exist, this aspect has received little attention in hydrology and hydrogeology. Comparative studies are rare, and methodological discussions are mostly lacking. An exception is Kraft et al. (2025), who evaluated three static–dynamic fusion methods as part of their hyperparameter tuning. However, this was a secondary aspect of their study, which primarily focused on reconstructing daily runoff.

Most existing studies adopt the simplest solution: replicating the static features at each time step to match the shape of the dynamic inputs, and then feeding both into a time series model such as a long short-term memory network (LSTM). One of the first to implement this approach were Kratzert et al. (2018), and since it yielded good results (and a later attempt to use static features as inputs to the static input gate of the LSTM performed worse (Kratzert et al., 2019)), many subsequent studies likely followed this strategy without further experimentation (Lees et al., 2021; Hashemi et al., 2022).

However, it seems intuitive that feeding time-invariant data directly into a time series model may not be optimal — or at the very least, not particularly efficient. An alternative approach, increasingly found in water-related studies, involves processing static features in a separate, non-sequential model component, such as a simple feed-forward network or multi-layer perceptron (MLP). The resulting representation is then concatenated with the output of the dynamic (time-series) model component — typically an LSTM — at a later stage (Heudorfer et al., 2023; Martel et al., 2024b, a; Arsenault et al., 2023).

Despite this emerging diversity, no study in the water sciences has, to our knowledge, systematically compared different strategies for integrating static features into global deep learning models. In contrast, the integration of dynamic and static inputs has been widely recognized as a methodological challenge in other disciplines, where it has been shown that the choice of integration strategy can significantly influence model performance.

For instance, Rahman et al. (2020) combined static user profile data (e.g., demographic attributes, preferences) with sequential dynamic input (design decisions over time) in a deep recurrent neural network to predict human design behavior. They explored different fusion strategies in a deep recurrent neural network, including early fusion (concatenation at the input level), mid-level fusion (after temporal encoding), and late fusion (combining representations just before output). Marx et al. (2023) explored blood glucose forecasting using both static patient information (e.g., age, disease duration, treatment plan) and dynamic time-series data (glucose levels). They used a modular architecture with parallel processing of static and dynamic inputs, followed by concatenation. Miebs et al. (2020) explored static feature integration strategies in the context of energy consumption forecasting, using short, regular time series (e.g., temperature, occupancy) alongside static building attributes (e.g., size, type, insulation). They systematically compared repetition, concatenation, conditional initialization, and feature-wise transformations of static inputs in RNNs. Liu et al. (2022) applied early fusion by concatenating static features with dynamic meteorological time series for crop yield prediction. Similarly, Wang et al. (2022) used concatenation of dense and LSTM-based encodings for predicting medical crowdfunding outcomes and found additional benefit from temporal attention in some cases.



The goal of this study is to systematically compare different approaches for processing dynamic and static input data within a global deep learning model for groundwater level prediction. To this end, we reviewed relevant literature from various disciplines and incorporated both the commonly used methods in water sciences as well as additional approaches that have shown promising results in other fields.

We used a subset of a recently published, machine-learning-ready long-term groundwater level dataset for Germany (Ohmer et al., 2025a, b). The subset includes 667 groundwater wells, selected from the full dataset based on their Nash–Sutcliffe Efficiency (NSE) in an initial single-well benchmark model. Only wells with an NSE > 0.7 were included, in order to exclude those clearly influenced by non-meteorological factors such as pumping. The dataset comprises groundwater level data, meteorological forcings and static environmental features for each well.

In addition, we conducted a parallel analysis using so-called time series features. i.e. statistical descriptors derived directly from the groundwater level time series (e.g., periodicity, seasonality) in place of the environmental static features provided by the dataset. This decision was motivated by previous findings showing that environmental static features, unlike time series features, were not well suited for spatial generalization (Heudorfer et al., 2023).

We tested four different approaches for integrating static features, each evaluated in both an in-sample (IS) setting (i.e., generalization in time) and an out-of-sample (OOS) setting (i.e., generalization in space and time), using both environmental and time-series-based static inputs. The aim was to investigate whether the method used to incorporate static features influences model performance, whether this effect differs between the IS and OOS settings, and whether the type and quality of static features impact the results.

In both settings, we also included a baseline model without static features. While in the IS setting, static features may serve primarily as identifiers or allow the model to benefit from training data associated with similar wells, in the OOS setting, static features are crucial for spatial generalization. They provide the only mechanism by which the model can infer that wells with similar static properties should respond similarly to identical dynamic inputs. By comparing the performance of models with and without static features — especially in the OOS setting — we assess to what extent each integration strategy is able to leverage static input features to improve generalization.

While time series features have shown superior performance in practice, they rely on past groundwater level observations and therefore do not allow for true spatial generalization to unmonitored sites. A trained model using these features can only be applied to wells where historical groundwater data are available — since such data are required to compute the features. In these cases, the wells could also be included directly in the training process, rendering spatial generalization unnecessary. In contrast, true spatial generalization in groundwater modeling is only possible when using spatially continuous environmental static features. In this study, we use time series features as a proxy for meaningful static inputs, because previous work has shown that commonly available environmental static features in groundwater applications are often not informative enough to support generalization (Heudorfer et al., 2023).



## 2 Data

### 2.1 Groundwater level data

We used a subset of a recently published, machine-learning-ready long-term groundwater level dataset for Germany (Ohmer et al., 2025a, b). The subset comprises 667 groundwater wells, each with a total record length of 32 years of weekly data from 1991-2022. Wells were selected from the full dataset based on their Nash–Sutcliffe Efficiency (NSE) in an initial single-well benchmark model. Only wells with an NSE > 0.7 were included, in order to exclude those that are clearly influenced by non-meteorological factors such as pumping. As a result, it can be assumed that the groundwater dynamics of the selected wells are

primarily driven by meteorological forcing. The wells are spatially well distributed across Germany (Figure 1) and represent a wide range of hydrogeological and climatic conditions (see provided time series plots in Zitat Zenodo).

### 2.2 Meteorological forcings as dynamic input data

The meteorological forcings were also obtained from the dataset mentioned above. They include weekly aggregated variables such as mean, maximum, and minimum air temperature, precipitation sum, and relative humidity, all derived from the HYRAS

dataset provided by the German Meteorological Service (DWD) (Razafimaharo et al., 2020). Additional variables — also from DWD — include real, potential, and reference evapotranspiration (FAO), as well as soil moisture and soil temperature at 5 cm depth (DWD-CDC, 2024). Further inputs, such as snow water equivalent, snowfall, snowmelt, and surface and subsurface runoff, were sourced from the ERA5-Land reanalysis dataset (Muñoz-Sabater et al., 2021).

### 2.3 Environmental features as static input data

The selection of static environmental features was based on the dataset (Ohmer et al., 2025a, b). These features include hydrogeological and soil characteristics (e.g., aquifer type, hydraulic conductivity, soil type, recharge), topographic attributes (elevation, slope, aspect, flow direction, etc.), and land use information.

From the full set of static features provided in the dataset Ohmer et al. (2025a, b), variables related to well depth, screen characteristics, pumping, and pressure state were excluded, as these were sparsely available for the majority of monitoring

wells. All categorical static features were label encoded for use in the machine learning models.

### 2.4 Time series features as static input data

Time series features are quantitative metrics derived from the groundwater level time series that describe specific aspects of their temporal dynamics. In this study, we use the feature set developed by Wunsch et al. (2022) and also applied in Heudorfer et al. (2023). This set consists of a redundancy-reduced selection of nine time series features, which has previously been used

successfully to cluster large sets of groundwater wells based on their dynamic behavior (Wunsch et al., 2022). For a complete list and detailed descriptions of the time series features used in this study, we refer to Heudorfer et al. (2023).



## 3  Methods

### 3.1  Long Short-Term Memory network

In all model setups, we used Long Short-Term Memory (LSTM) networks (Hochreiter and Schmidhuber, 1997) to process the
dynamic time series data. LSTMs are the most commonly applied machine learning architecture in water science, and as such,
we refrain from providing a detailed description here.

### 3.2  Incorporation of static input data

We implemented four different approaches to incorporate static input data into the models (Figure 2). The first two represent
the most widely used methods in water science to date, while the latter two were selected based on their promising performance
in comparative studies from other disciplines.

- *Repetition of static data*: The static features are replicated at each time step and fed into the Long Short-Term Memory
  (LSTM) network alongside the dynamic inputs, as first proposed in Kratzert et al. (2018). In the following, we refer to
  this architecture as the repetition model (rep model).

- *Concatenation of separately processed static data*: The static features are processed separately using fully connected
  layers, and their output is then concatenated with the final hidden state of the LSTM. The architecture corresponds to
  that used in (Ohmer et al., 2025a) and has previously been applied in studies such as Miebs et al. (2020) and Liu et al.
  (2022). We refer to this architecture as the concatenation model (conc model).

- *Attention mechanism on static features*: In this approach, the static features are first processed by a fully connected layer,
  and the resulting output is used to generate attention weights, which are then applied to the LSTM hidden states to
  compute a weighted sum as the final representation. This model was proposed by Liu et al. (2022), and was inspired by
  earlier work from Guo et al. (2019) and He et al. (2016). We refer to this as the attention model (att model).

- *Initialization of hidden and cell states using static features*: Here, the static features are first processed by fully connected
  layers, and the resulting output is used to initialize the hidden and cell states of the LSTM. This allows the static context
  to directly influence the dynamic sequence processing from the beginning. This method was employed by Miebs et al.
  (2020) and Liu et al. (2022), and in a slightly modified form by Wang et al. (2022). Following the terminology in those
  studies, we refer to this architecture as the conditional model (cond model).

### 3.3  Model Setup

#### 3.3.1  General Settings and Hyperparameters

All models were implemented with a single LSTM layer of size 128. The repetition model was initially tested in two config-
urations: one with an LSTM size of 128, to maintain consistency with the other architectures, and another with a size of 256,





to account for the increased input dimensionality resulting from the replication of static features at each time step. Throughout the results section, only the outcomes from the 256-neuron version are presented, as they yielded slightly better performance than the 128-neuron configuration.

In all model architectures, every neural layer — except for the output layer — is followed by a dropout layer with a dropout
rate of 0.3 for regularization. All models were trained using a batch size of 512 and a maximum of 20 epochs, combined with early stopping based on validation loss, with a patience of 5. A learning rate scheduling scheme was applied, targeting a final learning rate of 0.001. All models were trained using mean squared error (MSE) as the loss function.

The concatenation model includes a second model branch consisting of a Dense layer with 128 neurons to process the static input features. The outputs of this branch are concatenated with the final hidden state of the LSTM branch, followed by a Dense
layer of size 256 and a final Dense output layer with one neuron.

The attention model applies a static-driven attention mechanism where attention weights over the LSTM outputs are computed based on the static features, which are passed through a dense layer of size 128 before. The attention mechanism uses another dense layer to compute attention scores of size equal to the number of time steps. The resulting attention-weighted representation of the LSTM outputs is then passed through a dense layer of size 256, followed by a dropout layer and a final
linear output layer.

The conditional model features a second branch that processes the static features through a dense layer of size 128, followed by a dropout layer and another dense layer of size 2×LSTM units (i.e., 2×128 = 256). The output of this layer is split and used to initialize the hidden and cell states of the first LSTM layer in the dynamic branch. To improve temporal abstraction, we added a second LSTM layer with 128 units following the first, which outputs the full hidden sequence. The first LSTM is
initialized with static features (conditional setup), and the second LSTM summarizes the sequence into a single representation. We observed that this additional LSTM layer significantly improved model performance compared to a single-layer setup.

An overview of all model architectures is provided in Figure 2.

All models were evaluated on the final 10 years of the dataset, spanning the period from 2013 to 2022. The preceding years were used for training (1991–2007) and validation with early stopping (2008–2012). The input sequence length for the dynamic
inputs was set to 52 weeks (i.e., one year) for all models. Performance metrics were computed based on the median predictions of an ensemble of 10 model initializations.

### 3.3.2   In-sample Setting

In the in-sample (IS) setting, the models are trained on the training data from all wells, and predictions are made for the designated test period. This setting represents a generalization in time, as the model learns from each well and is evaluated on
future data from the same wells.

### 3.3.3   Out-of-sample Setting

The out-of-sample (OOS) setting was implemented as a 10-fold cross-validation (CV). The 667 wells were randomly divided into ten folds; in each run, the model was trained on the training period data from nine folds, and the tenth fold was held out



for testing — again using the same test period. This setup requires the model to generalize across both space and time, as it is evaluated on wells it has not seen during training.

## 4 Results and Discussion

### 4.1 Environmental static features

In the in-sample (IS) setting using static environmental features, the repetition model achieves the best performance across all error metrics, closely followed by the conditional and concatenation models (Table 1, Figure 3a). The median NSE is 0.81 for the repetition model, 0.80 for the conditional model, and 0.79 for the concatenation model. The attention model lags slightly behind, with a median NSE of 0.77. As expected, all models incorporating static features outperform the dynamic-only baseline, which yields a median NSE of 0.73.

A closer look at the results reveals that model performance often depends on the individual well. For the majority of wells, the results align with the overall trend: the repetition model performs best, followed by the conditional, concatenation, and attention models in that order. However, there are also wells that deviate from this pattern. This becomes evident when counting how many wells were best predicted by each model based on NSE, regardless of the margin.

Out of the 667 wells, 376 were best modeled using the repetition approach, followed by 109 wells for which the conditional model performed best. The concatenation model was optimal for 81 wells, and the attention model for 51 wells. Interestingly, 50 wells were best modeled by the dynamic-only (dynonly) model, suggesting that these wells did not benefit from the inclusion of static features.

We did not analyze this aspect in detail, as it would likely be impractical to deploy different architectures for different wells, even if a relationship to time series dynamics could be established. Nonetheless, this finding is noteworthy, and it would be valuable to investigate in future work whether this variation is merely coincidental or, more likely, as we suspect, related to the underlying temporal behavior of individual wells.

In the out-of-sample (OOS) setting, the performance of all models is significantly lower than in the IS setting, with median NSE values decreasing from 0.73–0.81 (IS) to 0.73–0.74 (OOS), as shown in Table 1 and Figure 3 b. The performance differences between the various approaches are minimal, and all models perform approximately on par with the dynamic-only baseline. This indicates that none of the models were able to effectively generalize in space using the available environmental static features.

The lower performance in the OOS setting, compared to the IS setting, can be attributed to three possible factors:

- **Reduced training data due to cross-validation:** In the 10-fold cross-validation setup, the amount of training data is reduced by 10 % in each fold. While this effect could theoretically be minimized using a leave-one-out cross-validation strategy — training the model on all but one well and testing on the remaining one — this would require $n$ separate model runs (where $n$ is the number of wells), making it computationally infeasible. Consequently, 10-fold CV remains a widely accepted and practical alternative.





– **Low quality of static features:** The environmental static features may lack sufficient quality, containing errors or uncertainties. This is likely, as many important variables influencing groundwater response to meteorological inputs — such as hydraulic conductivity — are often not measured directly at the well location. Instead, they are typically interpolated, estimated based on aquifer material, or modeled, and thus may not accurately reflect the true subsurface heterogeneity.

– **Non-representative well selection with respect to static features:** The set of wells used for training and testing may not be representative in terms of their static feature characteristics. If the variance in static properties is either too low or too high, the model may struggle to learn generalizable relationships, limiting its ability to extrapolate to unseen wells.

The latter two points are also likely explanations for the observation that models using environmental static features do not outperform the model without static inputs. This finding is consistent with the results of Heudorfer et al. (2023), who reported

similar outcomes using a much smaller dataset. Therefore, it appears increasingly likely that the limitation lies in the quality of the static features, rather than in the representativeness of the well selection.

## 4.2   Time series static features

As shown by Heudorfer et al. (2023), time-series features can lead to improved performance in the OOS setting. Based on this finding, we repeated all analyses using time-series features as static inputs, in order to evaluate whether and to what extent

the performance of the different integration approaches changes when meaningful static features are used. In this context, "meaningful" refers to features that enable the models to genuinely generalize across different wells based on the information provided.

In the in-sample (IS) setting with time-series features as static inputs, the results are particularly noteworthy. While the performance of the repetition model does not improve (median NSE: 0.81 vs. 0.81 previously), the concatenation and attention

models appear to benefit from the inclusion of meaningful static features (median NSE: 0.81, 0.80 vs. 0.79, 0.77 previously) (Table 1, Figure 3c). The conditional model performs slightly worse (median NSE: 0.79 vs. 0.80 previously).

Although the repetition model remains the top performer for 238 wells, the concatenation model catches up, now performing best for 198 wells. The attention model is best for 125 wells. The conditional model falls further behind, being optimal for only 60 wells, while the dynamic-only model remains best for 46 wells.

In the IS setting, models appear to benefit even from "meaningless" static inputs, likely by using them as a form of unique identifier (Heudorfer et al., 2023). However, when meaningful static features are provided — such as time-series-derived descriptors — the models gain the ability to generalize more effectively based on these inputs. This ability, however, is not equally pronounced across all integration strategies. While the concatenation and attention models show clear improvements with meaningful static features, the repetition model's performance remains largely unchanged, and the conditional model even

shows a slight decline.

In the out-of-sample (OOS) setting, as expected, all models show improved performance compared to when environmental static features were used. All integration approaches now outperform the dynamic-only model across all error metrics. In terms of median NSE, the repetition again performs best (0.82), closely followed by the concatenation model (0.80), conditional




model (0.79) and attention model (0.78). The dynamic-only model remains at a lower performance level, with a median NSE
of 0.73 (Table 1, Figure 3 d). Notably, the OOS results using time-series-based static features are only slightly lower than the
in-sample (IS) results, and in case of the repetition model even higher. These results confirm that the models are able to extract
meaningful information from time-series-based static features and use it to generalize across space.

When evaluating which model performs best for the highest number of wells, the repetition model takes the lead with 378
wells. It is followed by the concatenation model with 134 wells. The conditional and attention models lag behind, performing
best for only 64 and 58 wells, respectively. The dynamic-only model comes last, showing the best performance in just 33 wells.

## 4.3 Computational effort

In terms of computational effort, the different approaches exhibit noticible differences. While we were unable to consistently
track exact runtimes — due to parallel execution across machines with varying computational resources to reduce total run-
time — distinct trends emerged. The attention model was the fastest overall, closely followed by the concatenation model.
In contrast, both the repetition and conditional models were significantly slower, with the repetition model also demanding
considerably more memory.

## 4.4 Comparison with results from other disciplines

When comparing our findings to studies from other domains, several consistent patterns emerge regarding the value and inte-
gration of static features in deep learning models. However, the effectiveness of particular integration strategies also appears to
be domain-specific and strongly dependent on the nature and quality of the static features.

In general, studies across domains agree that meaningful static features can substantially improve predictive performance
and spatial generalization — but only when integrated using an appropriate strategy. Our results support this trend: when
switching from environmental to time-series-based static features, some of the tested architectures showed clear improvements
in both IS and OOS settings, while others remained unchanged or even declined.

In design science, Rahman et al. (2020) also showed that informed integration methods outperform simpler schemes. Among
these, mid-level fusion performed best, allowing the model to first encode temporal dependencies before incorporating static
context. This strategy corresponds most closely to our conditional architecture, which also achieved strong performance, albeit
not the highest in our experiments.

In medicine, Marx et al. (2023) found that parallel encoding and late fusion of static and dynamic features improved gener-
alization across patients. This setup is conceptually similar to our concatenation architecture. They reported that this approach
significantly improved performance, particularly for unseen patients, emphasizing that parallel processing followed by con-
catenation was a key to successful generalization. In our case, while the concatenation strategy also performed well under
out-of-sample conditions — especially when static features carried meaningful information — the repetition model still out-
performed it overall.

Miebs et al. (2020) found that both concatenation and conditional initialization strategies achieved the best trade-off between
predictive accuracy and computational efficiency. These findings are at least partially consistent with our results, in which both





the conditional and concatenation models also achieved strong performance. In particular, the concatenation model represents a favorable compromise between predictive accuracy and computational efficiency.

In agricultural modeling, Liu et al. (2022) compared models with and without static inputs and found that early fusion via concatenation of static and dynamic features yielded the best performance in both cross-validation and out-of-sample prediction. Their findings support our conclusion that concatenation is effective when static inputs are informative.

Wang et al. (2022) found that their best performance came from concatenating static and dynamic representations, with some benefit from temporal attention — which aligns with our finding that attention-based models can benefit from meaningful static features, though in our case, attention still slightly trailed behind concatenation and repetition.

Taken together, these studies confirm that the best-performing integration method depends not only on model architecture, but also on the quality and informativeness of the static features. Our work adds to this understanding by systematically evaluating several approaches in a large-scale groundwater modeling context and confirming that when static features are meaningful, concatenation offers an effective and efficient compromise, though with slightly lower predictive performance compared to the repetition model.

## 5 Conclusions

To address the central research question — to what extent does the choice of integration strategy for static features influence the performance of global deep learning models for groundwater level prediction? — the answer is: it depends.

First, the answer depends on whether the task involves an in-sample (IS) setting, i.e., generalization in time only, or an out-of-sample (OOS) setting, which requires generalization in both space and time. Second, it depends on the type and quality of static features available.

If no meaningful static features are available (as was the case with the environmental features used in this study), they can still improve performance in the IS setting — likely because they function as "unique identifiers" for each well (Heudorfer et al., 2023). However, under these conditions, performance differences among the various integration approaches are relatively minor, as long as static features are included at all. The repetition approach performs best in this setting, likely because it feeds the static features directly into the LSTM, allowing the model to effectively use them as unique identifiers. Nonetheless, the conditional and concatenation models offer viable alternatives: they achieve nearly the same performance while providing benefits such as faster training, greater stability, and lower memory usage.

None of the models demonstrated strong generalization when relying solely on the environmental static features in the OOS setting. We attribute this to the limited quality of these features. In such cases, the inclusion of low-quality static features provides no tangible benefit, regardless of the integration method.

When meaningful static features are used — in our case, the time-series features — the performance over all model approaches increases, particularly in the OOS setting, as expected. While the repetition model continues to achieve the highest performance, evaluation metrics again remain comparable across all approaches.





Overall, we conclude that all tested approaches for incorporating static features into global deep learning models perform
well, with only subtle differences in performance. It is important to note that we did not perform hyperparameter optimization
for any of the models. As a result, the current configurations may not be equally well-tuned across architectures, which could
influence the comparative performance to some extent. However, based on our previous experience, we expect any impact to be
limited, as hyperparameter tuning has typically led to only marginal improvements in model performance in similar settings.

More importantly, our results indicate that the quality of the static features has a greater impact on model performance than
320 the specific integration strategy, especially when it comes to out-of-sample predictions. While this may appear intuitive in
theory, it is often challenging to realize in practice — particularly in the context of groundwater modeling, where high-quality
static data are not always readily available.

Finally, when comparing our results to those from other disciplines, we find strong cross-disciplinary support for the con-
clusion that the optimal approach depends heavily on the amount, diversity (e.g., in terms of time series dynamics), represen-
325 tativeness, and quality of the available data — especially the static inputs. These factors vary not only between disciplines but
often also across datasets within the same field.

As a final remark, we note that the relatively simple repetition approach achieved consistently strong results in our study.
However, this method is not particularly efficient, as it involves replicating static features at every time step, which increases
memory consumption and computational cost. Depending on the specific characteristics of a given dataset and the available
resources, it may therefore be worthwhile to explore alternative integration strategies that offer a better balance between effi-
ciency and performance.

While our findings are directly applicable to groundwater level modeling with larger datasets, they may also be relevant to
related domains, such as surface water runoff prediction. However, they may not be universally transferable. A careful selection
of both, input features and integration strategy, remains essential to achieve the best possible model performance.

*Code and data availability.* The code used in this study is publicly available on GitHub (https://github.com/KITHydrogeology/dynamic_
static). All data used in this study are publicly available via Zenodo (https://https://orcid.org/0000-0001-8648-5333).

*Author contributions.* TL conceived the study, designed the experiments, and carried out all computations. MO contributed to discussions
on methodology and created the figures. TL wrote the original draft of the manuscript; MO reviewed and edited the text.

*Competing interests.* The authors declare that they have no conflict of interest



*Acknowledgements.* All programming was done in Python version 3.12 (van Rossum, 1995) and the associated libraries, including NumPy (Harris et al., 2020), Pandas (McKinney, 2010), Tensorflow (Abadi et al., 2016), Keras (Chollet et al., 2015), SciPy (Virtanen et al., 2020), Scikit-learn (Pedregosa et al., 2011) and Matplotlib (Hunter, 2007). The authors further acknowledge support by the state of Baden-Württemberg through bwHPC.



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



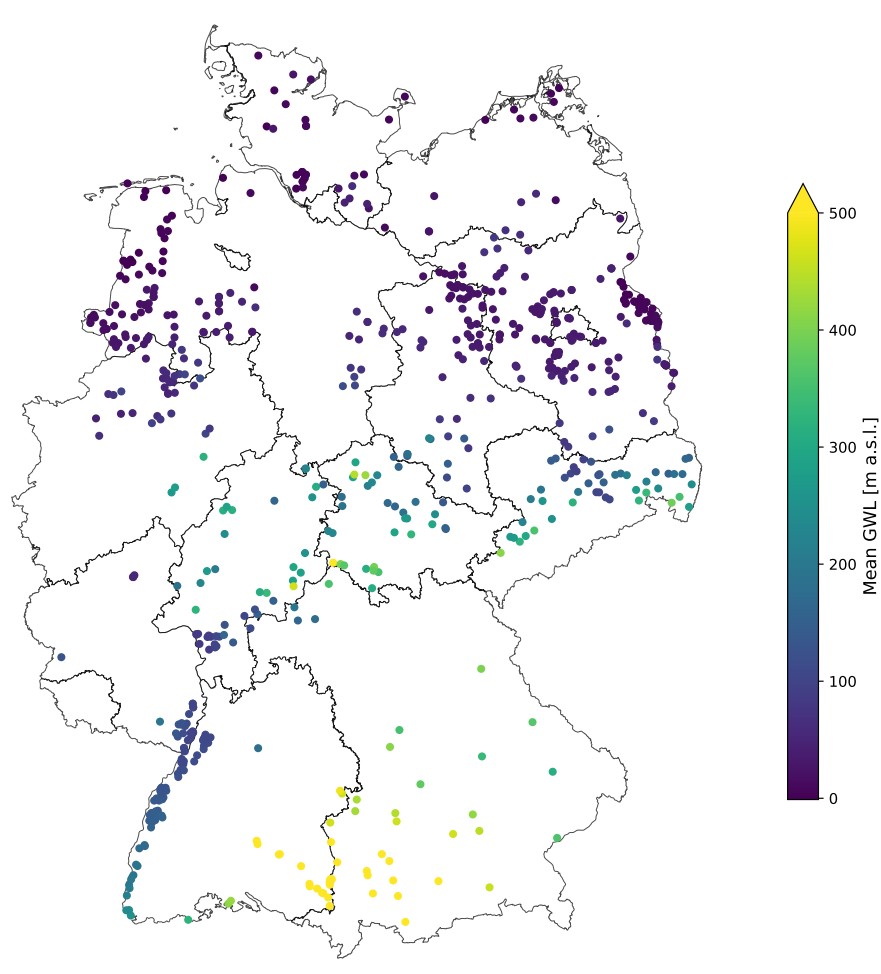

**Figure 1.** Location of the 667 selected groundwater wells, along with their mean groundwater level from 1991-2022.





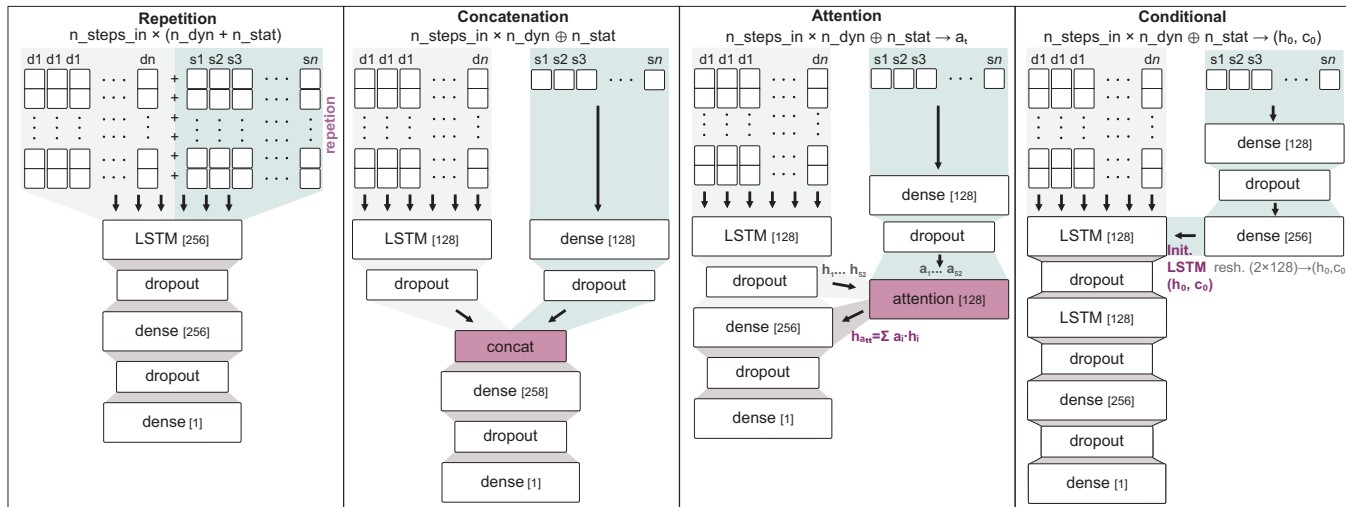

**Figure 2.** Model architectures used to incorporate static features into the global deep learning model.





**Figure 3.** Comparison of the NSE of all approaches as boxplots and cumulative distributions for the IS setting with environmental static features (a), OOS setting with environmental static features (b), IS setting with time series static features (c) and OOS setting with time series static features (d).





**Table 1.** Overview of metrics for all model approaches: In-sample (IS), out-of-sample (OOS), environmental static features (env), time series static features (ts), models without static features (dynonly), repetition model (rep), concatenation model (conc), attention model (att) and conditional model (cond).

| Model Approach | NSE | | | | RMSE | | | | R² | | | |
|---|---|---|---|---|---|---|---|---|---|---|---|---|
| | min | mean | med | max | min | mean | med | max | min | mean | med | max |
| IS dynonly | -2.32 | 0.69 | 0.73 | 0.92 | 0.05 | 0.32 | 0.19 | 6.27 | < 0.01 | 0.72 | 0.76 | 0.94 |
| IS env rep | -1.18 | 0.79 | 0.81 | 0.94 | 0.05 | 0.25 | 0.16 | 5.05 | < 0.01 | 0.82 | 0.84 | 0.96 |
| IS env conc | -1.00 | 0.77 | 0.79 | 0.93 | 0.06 | 0.27 | 0.17 | 4.84 | < 0.01 | 0.80 | 0.82 | 0.96 |
| IS env att | -1.93 | 0.74 | 0.77 | 0.94 | 0.06 | 0.29 | 0.18 | 5.86 | 0.03 | 0.77 | 0.80 | 0.95 |
| IS env cond | -1.50 | 0.77 | 0.80 | 0.93 | 0.05 | 0.27 | 0.17 | 5.41 | < 0.01 | 0.80 | 0.82 | 0.95 |
| OOS dynonly | -2.29 | 0.69 | 0.73 | 0.92 | 0.05 | 0.32 | 0.20 | 6.21 | 0.01 | 0.72 | 0.76 | 0.94 |
| OOS env rep | -1.57 | 0.69 | 0.74 | 0.93 | 0.06 | 0.31 | 0.19 | 6.29 | < 0.01 | 0.72 | 0.76 | 0.94 |
| OOS env conc | -1.85 | 0.69 | 0.74 | 0.93 | 0.06 | 0.32 | 0.19 | 6.17 | 0.01 | 0.72 | 0.76 | 0.95 |
| OOS env att | -2.17 | 0.69 | 0.73 | 0.94 | 0.06 | 0.32 | 0.19 | 6.74 | 0.02 | 0.73 | 0.76 | 0.96 |
| OOS env cond | -2.26 | 0.69 | 0.74 | 0.93 | 0.06 | 0.32 | 0.19 | 6.34 | 0.01 | 0.72 | 0.77 | 0.95 |
| IS dynonly | -2.37 | 0.69 | 0.73 | 0.92 | 0.05 | 0.32 | 0.19 | 6.27 | < 0.01 | 0.72 | 0.54 | 0.94 |
| IS ts rep | -1.28 | 0.78 | 0.81 | 0.95 | 0.05 | 0.26 | 0.16 | 5.16 | 0.05 | 0.63 | 0.81 | 0.96 |
| IS ts conc | -0.64 | 0.78 | 0.81 | 0.96 | 0.05 | 0.26 | 0.16 | 4.79 | 0.03 | 0.66 | 0.82 | 0.97 |
| IS ts att | -1.78 | 0.76 | 0.80 | 0.96 | 0.05 | 0.28 | 0.17 | 5.70 | 0.02 | 0.67 | 0.79 | 0.96 |
| IS ts cond | -0.25 | 0.77 | 0.79 | 0.95 | 0.05 | 0.27 | 0.17 | 4.68 | < 0.01 | 0.68 | 0.80 | 0.95 |
| OOS dynonly | -2.29 | 0.69 | 0.73 | 0.92 | 0.05 | 0.32 | 0.20 | 6.21 | 0.01 | 0.72 | 0.76 | 0.94 |
| OOS ts rep | 0.29 | 0.80 | 0.82 | 0.95 | 0.05 | 0.24 | 0.16 | 3.89 | < 0.01 | 0.82 | 0.85 | 0.96 |
| OOS ts conc | -0.74 | 0.77 | 0.80 | 0.96 | 0.05 | 0.27 | 0.16 | 5.18 | 0.01 | 0.81 | 0.83 | 0.96 |
| OOS ts att | -2.16 | 0.75 | 0.78 | 0.96 | 0.02 | 0.25 | 0.36 | 8.06 | 0.03 | 0.78 | 0.81 | 0.96 |
| OOS ts cond | -0.92 | 0.77 | 0.79 | 0.94 | 0.04 | 0.25 | 0.34 | 5.99 | 0.03 | 0.79 | 0.82 | 0.96 |