# Peer review of "Strategies for Incorporating Static Features into Global Deep Learning Models"

_EGUsphere, 2025_

## Author Comment (AC2)

The authors compare four different strategies for incorporating static features into global deep learning models. They show that the repetition method generally achieves the best performance but they are less computationally efficient. They also state that the selection of static features is more important than the choice of integration strategies. The manuscript is generally well-presented.

We thank the reviewer for the positive assessment of the manuscript and for the constructive comments, which we believe help to further improve the quality and clarity of the paper. In the following, we address each point raised by the reviewer in detail.

However, I still have the following concerns.

In the Table 1 of Heudorfer et al. (2023), it shows that the time series features were derived from past groundwater level time series until 2011. The training, validation, and test periods for this study are 1991-2007, 2008-2012, 2013-2022, respectively. If the authors directly adopt the time series features from Heudorfer et al. (2023), there might be data leaking issue during validation.

We thank the reviewer for pointing this out. Although the feature definitions follow Heudorfer et al. (2023), all time-series-based static features were recomputed in this study using only information available up to the respective training period. In particular, no information from the validation or test periods was used in the feature derivation, thus preventing any data leakage. We will clarify this explicitly in the revised version of the manuscript.

Also, while the authors have provided references for the time series features, it is nice to list the time series features in the main text or appendix for readability.

Thank you for that constructive suggestion! For improved readability, we will add a complete list of the time-series-based static features to the Appendix.

For the conditional model, it is unclear to me how the output of this layer is split and used to initialize the hidden and cell states of the first LSTM layer in the dynamic branch (Line 167-168). Does it mean that the output is directly used as the initial condition of the hidden and cell states? Please clarify.

Thank you for the comment. We agree that this was not sufficiently clear. Yes— the static branch outputs a vector of length $2H$, which is split into two tensors of length $H$ and directly provided as the initial hidden and cell states (initial_state=[h0, c0]) of the first LSTM layer in the dynamic branch. We will clarify this in the revised manuscript.

In Line 176, the authors mentioned that there are 10 model initializations. Does it mean that the authors train 10 global models with different initial conditions?

Yes, correct. We will clarify that the ensemble consists of 10 independently trained global models with identical architecture, data splits, and hyperparameters, differing only in their random weight initialization.

For the results, the authors that the repetition model performs the best, but they show the results for the 256-neuron model for the repetition model and the 128-neuron model for the other models. It is hard to identify whether the better performance is due to the integration strategy or the more hidden neurons.

We thank the reviewer for the comment. As described in the manuscript, all models were implemented with a baseline LSTM size of 128 units. For the repetition strategy, we additionally tested a configuration with 256 units to account for the increased input dimensionality resulting from the replication of static features at each time step. Since this larger configuration yielded slightly

better performance, only the results of the 256-unit repetition model are reported in the Results section. We hope that this description sufficiently clarifies this point.

Specific comments:

Explain labels/legends in Figures 2 and 3.

Thank you for pointing this out. We will improve the figure captions and legends to explicitly explain all labels and symbols in Figures 2 and 3 in the revised manuscript.

---

## Author Comment (AC3)

This paper explores different strategies for global deep learning models that account for basin and hydrogeological "static" properties and characteristics. These strategies aim to enhance the models' generalization capabilities and overall performance. The authors tested several approaches that differ in how static properties are incorporated into the model and conducted these tests using two types of modeling methods ("in-sample"/training on all available wells, and "out-of-sample"/training 90% of the wells with test on the remaining 10%, test period being equal in both approaches). From the Deep Learning point of view, their study builds on this same technical issue that also emerged from other scientific fields. 4 integration strategies were tested for each modeling approach (in-sample and out-of-sample). The simplest integration strategy (repetition) appeared to perform the best, at the cost of much lower computational efficiency. However, the authors conclude that other strategies, particularly concatenation and conditional initialization of LSTM weights, deserve thorough consideration as they offer a good balance of performance and computational efficiency. The paper addresses an important issue; it summarizes the usefulness and relevance of existing strategies for incorporation of static features in LSTM in the specific case of hydrogeology. It is a very nice study, clearly written and organized. There are a few points that might be addressed or highlighted in the paper before publication in my opinion.

We thank the reviewer for the thorough and insightful review, as well as for the very positive assessment of the manuscript. We particularly appreciate the constructive comments and suggestions, which we believe help to further strengthen the clarity and relevance of the paper. In the following, we address each point raised by the reviewer in detail.

The paper should provide examples of time series (e.g. in the form of a panel with 4 or 5 of them) without requiring the reader to download them. I think it is important to have a straightforward understanding of the context of hydrological modeling by knowing what ground-truth data looks like. Therefore, it is crucial that a few examples be presented directly in the text. For instance, if most time series consist of almost pure periodic annual variations with constant amplitude through time, expectations regarding the model's performance would not necessarily be the same as for more complex variability. After downloading the time series and briefly examining them, significant differences in statistical properties can be observed (more or less weak trends, very short-term variations, strong amplitude of the water year cyclicity...). Do the authors know if, and to what extent, such differences may play any role in the models' performance: are there some behaviors for which the models systematically perform poorly, or very well?

We thank the reviewer for this insightful comment highlighting the importance of providing a direct impression of the groundwater level time series used in this study. To better illustrate the characteristics of the observed data, we will add a new figure to the manuscript showing representative examples of groundwater level time series from several wells, covering a range of typical behaviours such as pronounced seasonal cycles, long-term trends, and varying short-term variability.

Regarding the potential influence of different time series characteristics on model performance, we agree that this is an important and interesting question. In the present study, however, we did not conduct a systematic stratification of model performance by time series type or statistical properties. Our primary objective was to evaluate integration strategies for global models intended to operate consistently across large and heterogeneous monitoring networks. In such practical settings, it is typically not feasible to apply different modelling strategies depending on the behaviour of individual monitoring sites. Accordingly, while substantial heterogeneity in groundwater dynamics is present across wells, we think that a detailed performance analysis conditioned on specific time series characteristics is beyond the scope of this work and represents a promising direction for future research.

Although it is beyond the scope of the paper, it seems like a lot of meteorological inputs was used. To what extent are they all "meaningful" for the application? Have previous studies that used this database conducted SHAP analysis or similar methods to determine which features such models learn from most effectively?

We thank the reviewer for this relevant comment. In this study, we directly used the meteorological input variables provided by the GEMS-GER dataset without performing an explicit feature selection or feature attribution analysis. Our primary objective was not to assess the relevance of individual meteorological predictors, but to investigate how different strategies for integrating static features affect model performance and generalization.

We fully agree that analyzing which meteorological or static inputs are most informative—using methods such as SHAP or related feature attribution techniques—is a highly interesting and important topic. However, such analyses are beyond the scope of the present work and would substantially extend its focus. We therefore consider this a promising direction for future research.

I think one or two lines on the concept of "meaningful" static features as it is used here would be needed. Here, "meaningful" stands for "informative" if I am not mistaken; maybe this term would be more appropriate.

We thank the reviewer for this helpful suggestion. Following the reviewer's recommendation, we will replace the term "meaningful" with "informative" throughout the manuscript to improve clarity and precision.

It should be said at the beginning and justified why no hyperparameter optimization was conducted: this is a technical context that should be mentioned and explained (even briefly), especially for researchers who intend to use a similar approach.

We thank the reviewer for this important comment. In this study, we deliberately did not perform an extensive hyperparameter optimization, as our primary objective was to compare different integration strategies under consistent and comparable model configurations rather than to maximize predictive performance. All models therefore share the same baseline architecture and hyperparameter settings, allowing us to isolate the effects of the integration strategies. We will clarify this rationale in the manuscript.

Would one-hot encoding be very different than the repetition approach? This is the simplest way to attach an identifier to the wells, so in the framework of this study it would be interesting to recall this.

We thank the reviewer for raising this interesting point. One-hot encoding is indeed a simple and commonly used way to incorporate site identifiers as static inputs. Upon reflection, we note that one-hot encoding and the repetition strategy address different aspects of the model input: while one-hot encoding increases the dimensionality of the static feature space by encoding site identity, the repetition strategy operates along the temporal dimension by replicating static features at each time step of the dynamic input.

In this study, our focus was on integrating physically or environmentally informative static features rather than providing an explicit site identifier. Moreover, the set of static attributes used here is relatively rich (more than 40 variables) and may already provide an implicit characterization of individual wells, i.e., a "soft" identifier, while still being interpretable in terms of hydrogeological and environmental properties. We therefore did not include an additional one-hot well identifier as a

separate strategy. We agree, however, that an explicit comparison between identifier-based encodings (e.g., one-hot) and feature-based static descriptions would be an interesting direction for future work.

Line 149 (LSTM layer size): Why is it 128 when no hyperparameter optimization has been performed? Given that the results of the repetition model were presented for an LSTM layer of size 256, wouldn't it be preferable to present all results with an LSTM size of 256? I am not questioning the relevance of the results presented here, but it is important in my opinion that the presentation of the methodology does not raise any unnecessary questions for researchers interested in developing a similar approach.

As described in the manuscript, all models were implemented using a baseline LSTM size of 128 units in order to ensure consistency across integration strategies. For the repetition strategy, we additionally evaluated a configuration with 256 units to account for the increased input dimensionality resulting from the replication of static features at each time step. In this case, the number of dynamic input features is more than doubled compared to the other strategies, making a larger recurrent layer a reasonable and fair architectural choice, even in the absence of extensive hyperparameter optimization. Since this larger configuration yielded slightly better performance, only the results of the 256-unit repetition model are reported. We hope that this explanation in the manuscript sufficiently clarifies the rationale behind the chosen configurations.

Line 106: replace "real" with "actual".

Thank you, we will replace the term "real" with "actual" in line 106.

Line 177: Section 3.3.2 and 3.3.3 should be merged in a single "In-sample and Out-of-sample" 3.3.2 section.

We thank the reviewer for this suggestion. We deliberately chose to describe the in-sample and out-of-sample settings in separate sections, as they differ in terms of training setup and evaluation objective. We believe that this separation improves the clarity of the experimental design and therefore decided to retain the current structure. We hope that this organization is acceptable to the reviewer.

About the quality of environmental static features:

Line 220: I am not sure I understand this point well, which also seems to be about the heterogeneity of physical characteristics, as discussed in the previous point. Regarding the representativeness of the database for sampling general hydrogeological characteristics properly, I believe this can be addressed with general hydrogeological knowledge. Additionally, the spatial coverage and number of wells appear sufficient for consistent sampling of hydrogeological properties.

We thank the reviewer for this thoughtful comment and the opportunity to clarify this point. We agree that two related, but distinct aspects are discussed in the manuscript, and we appreciate the reviewer's careful reading.

The first point concerns the representativeness of the selected monitoring sites with respect to static features. While we agree that the overall number of wells and their spatial coverage may be sufficient to sample general hydrogeological characteristics, the wells included in this study were selected based on data availability and model performance considerations, not on the representativeness of their static environmental attributes. We did not explicitly assess, either a priori or a posteriori, whether the selected wells adequately cover the full range of static feature variability present in the broader dataset. Consequently, it cannot be excluded that the training and test sets underrepresent

certain combinations or extremes of static environmental characteristics, which may limit the model's ability to generalize to unseen wells.

The second point relates to the quality of the static environmental features themselves. By "low-quality" static features, we do not imply erroneous data, but rather refer to limitations common to large-scale environmental datasets, such as coarse spatial resolution, interpolation, or indirect estimation of key subsurface properties. These limitations are distinct from, but can compound, issues of representativeness in the selection of monitoring sites.

Taken together, our results suggest that both the representativeness of static feature distributions across wells and the intrinsic quality and process relevance of the static features play an important role in determining model performance. A systematic assessment of static feature representativeness and targeted selection strategies based on hydrogeological criteria would therefore be a valuable direction for future research, but was beyond the scope of the present study.

Overall, I don't quite understand how one could conclude that poor-quality static features have a greater impact on model performance than the integration strategy. Does this mean that all the used static features were poor quality? Or are these features considered poor quality compared to static features derived from time series, which are all meaningful? It will always be very difficult (not to say almost impossible sometimes) to have all at once high-quality, large-scale, high-resolution hydrogeological characteristics data that precisely accounts for spatial heterogeneity. Are we then reaching a major limitation to improve even more the generalization capabilities of Deep Learning models? In that case, would there be some particularly crucial environmental static features to focus on?

We thank the reviewer for this important and nuanced comment, which highlights the need to clarify the interpretation of our conclusions. Our statement that static feature quality may have a greater impact on model performance than the integration strategy should be understood in a relative, not absolute, sense.

We do not imply that all static features used in this study are of poor quality, nor that integration strategies are unimportant. Rather, our results indicate that, within the range of integration strategies tested, differences in model performance were more strongly influenced by whether the static features were informative for the prediction task than by how these features were technically integrated into the model. In particular, time-series-derived static features—which directly summarize aspects of groundwater dynamics—proved to be consistently informative, whereas large-scale environmental static features are inherently limited by spatial resolution, heterogeneity, and indirect estimation.

We fully agree with the reviewer that obtaining high-quality, spatially consistent, and process-relevant hydrogeological characteristics at large scales is extremely challenging. We therefore do not view this as a fundamental limitation of deep learning approaches, but rather as an indication that future improvements in generalization performance are likely to depend more on advances in the availability, representativeness, and process relevance of static features than on further refinements of model architectures alone.

That said, identifying which environmental static attributes are particularly beneficial for large-scale groundwater prediction—such as features that more directly reflect aquifer properties or hydrological connectivity—represents an important and open direction for future research.